# Polymer-coated urea effects on yield and gaseous loss of nitrogen for planting Chinese cabbage

Lian Xie[1], Yifei Du[1], Li Tang[1¤a], Xujiao He[2¤b], Wei Ming[3¤c], Xiangmin Rong[2], Yong Xie[1*]

1 College of Agriculture, Guizhou University, Guiyang, China, 2 College of Resources, Hunan Agricultural University, Changsha, China, 3 Department of Agriculture and Rural Affairs of Guizhou Province, Guiyang, China

☯ These authors contributed equally to this work.
¤a Current Address: Jiaxiu South Road, Huaxi District, Guiyang City, Guizhou Province
¤b Current Address: No.1 Nongda Road, Furong District, Changsha City, Hunan Province
¤c Current Address: No. 62 Yan'an Middle Road, Guiyang City, Guizhou Province
* yongxie@gzu.edu.cn

## Abstract

The overuse of nitrogen fertilizer in Chinese vegetable fields, while boosting yield, is a major source of environmental pollution, particularly through $N_2O$ and $NH_3$ emissions. Optimizing nitrogen management is therefore crucial for reconciling productivity with environmental sustainability. Here, we assessed whether polymer-coated urea (PCU) with reduced application rates could sustain yield while minimizing environmental costs in Chinese cabbage production. Compared to conventional urea (260 kg N ha$^{-1}$), PCU applications reduced by 0–20% increased nitrogen use efficiency by 47.7–49.9% and did not affect yield, whereas a 30% reduction caused significant yield loss. These PCU treatments also significantly reduced cumulative $N_2O$ and $NH_3$ emissions by 43.5–52.7% and 20.7–40.9%, respectively. We therefore recommend a 20% reduction in PCU-N application as the optimal management strategy for sustainable Chinese cabbage production, ensuring high yield with minimal environmental impact.

## Introduction

Global anthropogenic emissions of nitrous oxide ($N_2O$) and ammonia ($NH_3$) have increased significantly over the last 30 years, triggering serious environmental problems [1]. $N_2O$ is a potent greenhouse gas (GHG), with a 100-year global warming potential that is 300 times higher than carbon dioxide ($CO_2$) [2]. Moreover, it has become the predominant ozone-depleting substance in the 21st century [3]. The deposition of $NH_3$ is a serious threat to the environment and public health through the induction of acidification, eutrophication, and indirect $N_2O$ emission [2,4]. It can also facilitate the formation of $PM_{2.5}$ (fine particles smaller than 2.5 μm) by acting as

**Data availability statement:** All relevant data are within the manuscript and its Supporting Information files.

**Funding:** This study was financially supported by S&T Plan Project of Guizhou Province[grant number 黔科合基础-ZK(2024)- General Project 029], Guizhou University Doctoral Fund [grant number 贵大人基合字(2022) 35] and Basic Research Projects of Guizhou University[grant number 贵大基础(2023) 37].

**Competing interests:** NO authors have competing interests Enter: The authors have declared that no competing interests exist.

a precursor of secondary inorganic aerosols [5]. Not only does excessive and inappropriate use of nitrogen fertilizers in agriculture significantly reduce crop yields and economic benefits [6], but it also contributes significantly to anthropogenic emissions of $N_2O$ and $NH_3$ [7]. Among the various agroecosystems, intensive vegetable systems are among the most important sources of gaseous nitrogen, accounting for 9% and 12% of the world's $N_2O$ and $NH_3$ emissions, respectively [8,9]. Therefore, maintaining crop yield while reducing ammonia and greenhouse gas emissions is the top priority for sustainable agricultural production.

China is the world's leading vegetable producer, contributing approximately 40% of global production [10]. Unlike cereal crops, which often have extensive root systems that can efficiently explore deeper soil layers for nutrients, leafy vegetables such as *pakchoi* are characterized by rapid biomass accumulation and short growth cycles, which drive intensive nitrogen demands to meet yield targets within constrained cultivation periods [2]. While some vegetable crops may have smaller root systems, this does not necessarily equate to low nitrogen use efficiency (NUE) [11]. However, driven by economic interests and the pursuit of higher yield, intensive Chinese vegetable production has been characterized by high N fertilizer application rates, and the data showed that the average application of N fertilizer for vegetable production in China was 352 kg N ha$^{-1}$, twice the amount applied to vegetable cultivation in the United States [12]. These physiological and economic factors render China's vegetables accounting for 13% of the country's planting area, and they consume 25% of the country's chemical fertilizer while emitting 35% of crop-sourced greenhouse gases [13]. Thus, optimizing N management in vegetable production in China is an urgent issue not only for crop yield but also for the mitigation of greenhouse gas emissions in China.

N application significantly influences $N_2O$ and $NH_3$ emissions by affecting the main drivers of these emissions: N availability, soil pH, and microbial activity [14,15]. N fertilizer sources also play an important role in determining the amount of nitrogen loss by controlling the release rate of nitrogen, the transformation of microbial nitrogen, and the pH of the soil [16]. Consequently, it is well-established that $N_2O$ and $NH_3$ emissions in cropland increased with N application rate [17–19]. Therefore, reducing the N application rate is an effective method to control the loss of gas nitrogen [20]. For example, Zhang et al. (2021) demonstrated that the optimal N ratio could reduce overall GHG emissions by 18% from open-field vegetable production systems [21]. Similarly, Liang et al. (2020) found that long-term nitrogen (N) reduction in vegetable systems could decrease $N_2O$ and $NH_3$ emissions by 56.8% and 83.2%, respectively [22]. In this context, Polymer-Coated Urea (PCU) is considered one of the most effective controlled-release urea (CRU) owing to its superior nutrient-release properties [23]. PCU formulations using polyurethane, degradable polymers, and water-based coatings have seen widespread adoption in recent years due to their environmental safety, cost-effectiveness, and non-toxic properties [24]. While PCU carries a higher cost than conventional urea, its enhanced NUE can offset this premium. Studies demonstrate that PCU's yield advantages over conventional urea at equivalent N application rates in rice [25], wheat [26], and maize [27].

In China, PCU studies have predominantly focused on wheat and maize [28] growing areas in the north, and rice [29] growing areas in the south. Geng et al. [28] reported the CRU treatments significantly increased the yields of wheat and corn by 8–12% and 9–11%, respectively. A previous study reported that replacing conventional fertilizer with CRU at the same nitrogen dosage increased rice yield and NUE by 5.24% and 20.18%, respectively, while reducing $N_2O$ and $NH_3$ volatilization by 25.64% and 35.88% [29]. However, few studies have investigated the concurrent effects of PCU application on enhancing NUE and mitigating environmental risks in Chinese cabbage (*pakchoi*) production systems, particularly in the subtropical hilly drylands of southern China. Thus, the main purpose of this study was to evaluate the potential of PCU with reduced N rates in increasing crop yields and reducing the loss of gaseous N compared to conventional N practices in a subtropical hilly dryland system of southern China. We hypothesized that an optimal nitrogen management strategy, balancing crop productivity and environmental sustainability, could be identified for the current agricultural context in this region.

## Materials and methods

### Field experiment description

A field experiment was conducted in 2024 at the Liuyang Research Station, Changsha, China (28°19′N, 113°79′E). The soil was a light-loamy fluvo-aquic soil, which developed from alluvial sediments. The basic soil properties were as follows: pH of 5.8, SOM of 14.6 g kg$^{-1}$, total N of 1.0 g kg$^{-1}$, available N of 49.3 mg kg$^{-1}$, available P content of 11.8 mg kg$^{-1}$, and available K content of 157.1 mg kg$^{-1}$ in the 0–20 cm depth before the start of the experiment.

The experiment employed a randomized complete block design with three replicates. Six applied-N treatments were implemented: (1) **CK** (no N fertilizer); (2) **CU**: conventional urea (46% N) at 260 kg N ha$^{-1}$, applied following local practice (50% basal, 50% topdressed at 3 weeks after transplanting); (3) **1PCU**: polyurethane-coated urea (PCU; 45% N, 45-day release period; Shandong Nongda Fertilizer Technology Co., Ltd.) at 260 kg N ha$^{-1}$; (4) **0.9PCU**: PCU at 234 kg N ha$^{-1}$; (5) **0.8PCU**: PCU at 208 kg N ha$^{-1}$; (6) **0.7PCU**: PCU at 182 kg N ha$^{-1}$. In all PCU treatments, a one-off basal application of nitrogen fertilizer was carried out. The PCU used was characterized by its blue color, high size uniformity (99% of particles between 2–4.75 mm), and an initial nutrient release rate of ≤ 5% (S1 Fig). All plots received a uniform basal application of phosphorus and potassium at rates of 120 kg $P_2O_5$ ha$^{-1}$ (as calcium superphosphate; Xiangyun Chemical Industry Co. Ltd.) and 160 kg $K_2O$ ha$^{-1}$ (as potassium chloride; Uralkali), respectively. A common local cultivar of Chinese cabbage was grown in 4 m × 5 m plots. The crop was transplanted on October 10 and harvested on November 30, 2024, corresponding to a 50-day growth period. The planting density was $8 \times 10^4$ plants ha$^{-1}$, with row and plant spacing set at 0.3 m and 0.2 m, respectively. Throughout the growing season, the crop relied solely on natural precipitation, with meteorological conditions recorded in Fig 1.

### Yield and N efficiency

Ten cabbage plants were randomly collected from each plot during the crop harvesting period. The fresh cabbage plants sampled at the mature stage were weighed for cabbage yield. All cabbage plants were dried in an oven at 70°C for more than 48 hours and then weighed for the aboveground dry matter yield. The N content of the aboveground cabbage biomass was determined using the Kjeldahl digestion method. The N uptake of the cabbage plants was obtained by the following equation:

$$NUP = NC \times DMY \tag{1}$$

where *NUP* is the N uptake of the cabbage plants (kg N ha$^{-1}$), *NC* is the N content of aboveground biomass (%), and *DMY* is the aboveground dry matter yield (kg ha$^{-1}$). The NUE was calculated according to the equation below:

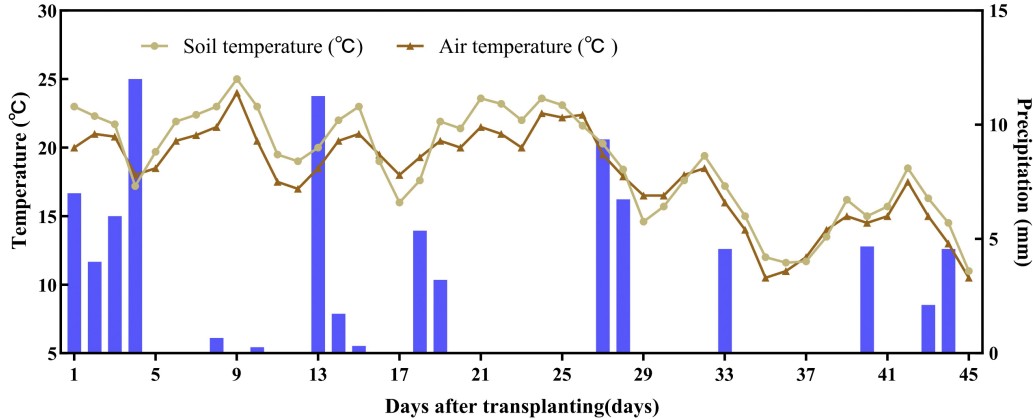

**Fig 1. Variations of soil temperature, air temperature and precipitation during cabbage planting in 2024.**

$$NUE = \frac{NUP_F - NUP_C}{N_{rate}}$$

(2)

where $NUE$ is the N use efficiency (%), $NUP_F$ and $NUP_C$ are the N uptake of cabbage plants (kg N ha$^{-1}$) under N fertilization and control treatments, respectively, and $N_{rate}$ is the N application rate (kg N ha$^{-1}$) for each strategy.

## Gaseous-N sampling and measurements

The $N_2O$ emissions were quantified using static chambers coupled with gas chromatography. Each chamber comprised a stainless steel base (50×50 cm) and a removable top (50×50×60 cm), with the base inserted 10 cm into the soil. Gas samples were collected from the chamber headspace at 10-minute intervals between 9:00 and 11:00 AM using a 20-mL syringe and stored in vacuum vials. $N_2O$ concentrations were analyzed using a Clarus 580 gas chromatograph (PerkinElmer, Waltham, MA, USA) equipped with an electron capture detector. The chamber air temperature was monitored during sampling. Measurements were conducted on days 1, 2, 3, 5, and 7 post-fertilization, followed by weekly sampling. Chamber recovery rates (98–101%) were validated using $N_2O$ standards.

$NH_3$ volatilization was measured via a continuous airflow enclosure system. Ambient air, pre-scrubbed of $NH_3$ by using boric acid, was drawn through a cylindrical chamber (19 cm diameter × 15 cm height) at 1.5 L min$^{-1}$ for 2 hours. The chamber was positioned 10 cm above the soil surface and volatilized $NH_3$ was trapped in 50 mL of 2% boric acid containing a mixed indicator (methyl red and bromocresol green). The trapped $NH_3$ was quantified by acid titration. Sampling occurred daily for 10 days after post-fertilization, then weekly, with measurements taken twice daily (9:00–11:00 AM and 3:00–5:00 PM). Fluxes were calculated from flow rates and collection times, and cumulative emissions were derived by summing daily rates.

## Emission factors

The emission ratio (ER) of $N_2O$ as $N_2O$-N for whole growing season was calculated as:

$$ER = \frac{(Total\ N_2O - N)_{N-applied} - (Total\ N_2O - N)_{Control}}{Total\ N\ applied} \times 100$$

(3)

where $(Total\ N_2O$-$N)_{N-applied}$ is the total $N_2O$-N emission for a synthetic N source treatment, $(Total\ N_2O$-$N)_{Control}$ is the total $N_2O$-N emission of the control treatment and $Total\ N\ applied$ is the application rate of N fertilizer (kg N ha$^{-1}$). The $ER$ for $NH_3$-N has been computed similarly.

## Economic benefit

The yield economic benefit (YEB, CNY ha$^{-1}$) was calculated as:

$$YEB = Y \times r \tag{4}$$

where $Y$ (kg ha$^{-1}$) refers to cabbage yield in the CU or PCU plots, and $r$ refers to the market price of the cabbage (1.5 CNY kg$^{-1}$).

The cost of nitrogen input was calculated as:

$$N\ input\ cost = N \times p \tag{5}$$

where $N$ refers to N rate (kg ha$^{-1}$), and $p$ refers to the market price of the nitrogen fertilizer of the CU (2.5 CNY kg$^{-1}$) or PCU (4.5 CNY kg$^{-1}$).

The environmental cost (CNY ha$^{-1}$) for each plot was calculated as:

$$Environmental\ cost = \sum_{a=1}^{b} Ca \times Pa \times m \tag{6}$$

where $Ca$ (kg N ha$^{-1}$) denotes the cumulate of 'a' (N$_2$O–N or NH$_3$–N) loss and $Pa$ (EUR kg N$^{-1}$) denotes the unit price of 'a' (N$_2$O–N or NH$_3$–N) loss to the environment that triggers adverse health, ecosystem and climate conditions, with fluctuations of 4–30 EUR kg N$^{-1}$ for NH$_3$-N to air, and of 6–18 EUR kg N$^{-1}$ for N$_2$O–N to air [30]. In addition, the value of $m$ is 7.6, which is the exchange rate of Euro to RMB.

The net economic benefit (NEB, CNY ha$^{-1}$) was calculated as:

$$NEB = YEB - N\ input\ cost - Environmental\ cost \tag{7}$$

where $YEB$ and $N\ input\ cost$ were calculated from the above. The highest net economic benefit was derived from the lowest environmental cost, while the lowest net economic benefit was derived from the highest environmental cost.

## Statistical analysis

All data (S2 Dataset) were drawn with Excel 2016 (Microsoft Corp., Redmond, WA, USA). Statistical analysis of the data was conducted with SPSS, version 19.0 (IBM Corp., Armonk, NY, USA). A one-way analysis of variance (ANOVA) was used to determine significant differences in the measured variables. Lowercase and uppercase letters in the figures and tables indicate statistically significant differences after Duncan's new multiple range test at $P < 0.05$ and $P < 0.01$.

## Results

### Temperature and precipitation

The cumulative precipitation was 83.81 mm during the Chinese cabbage growing season. The precipitation during the basal and topdressing fertilization stages was 51.85 mm and 32 mm, respectively (Fig 1). In general, changes in soil temperature and air temperature during the growing season of Chinese cabbage are basically consistent, both of which decrease slowly. The overall average air temperature and soil temperature were 20.8°C and 21.4°C, respectively. The average air and soil temperature in the basal and topdressing fertilization periods were 22.0°C, 19.5°C, 23.0°C and 19.8°C, respectively.

### Chinese cabbage yield

Cabbage yield under the CU fertilization treatment was not significantly greater than that under 1PCU, 0.9PCU, and 0.8PCU treatments, whereas significant differences were observed between the CU fertilization treatment and 0.7PCU

treatment (Fig 2). Compared with the cabbage yield of 23196.67 kg ha $^{-1}$ in the CU, increasing the yield by 7.6% for the 1PCU, 4.1% for the 0.9PCU, while decreasing the yield by 13.8% for the 0.8PCU, 26.6% for the 0.7PCU. There were no significant differences between 0.8PCU and 0.7 PCU.

### N uptake and utilization

Compared to the CU plot, N uptake was significantly improved by 21.9% for the 1PCU, 17.0% for the 0.9PCU, slightly increased for the 0.8PCU, and decreased by 8.5% for the 0.7PCU ($P > 0.05$). There were no significant differences in N uptake among the PCU treatments that reduced 0–20% N application (Table 1). However, the N uptake of 0.7PCU was significantly lower than that of other PCU treatments. There was significant variation between CU and reductions of 0–20% in PCU-N application for the NUE, while there were no significant differences between CU and 0.7PCU treatment. Relative to the CU plots, NUE was improved by 45.4% for the 1PCU, by 50.3% for the 0.9PCU, by 54.0% for the 0.8PCU, and by 1.5% for the 0.7PCU. There was significant variation between 0.7PCU and other PCU treatments.

### N$_2$O emission flux

There were obvious temporal changes in the N$_2$O emissions among the different fertilizer treatments during the Chinese cabbage growing season (Fig 3). Significant N$_2$O emissions were shown in the CU plot about four days after N application, with the peak fluxes reaching 1.21 mg N m$^{-2}$ h$^{-1}$ for the basal fertilizer stage and 1.13 mg N m$^{-2}$ h$^{-1}$ for the topdressing stage. However, N$_2$O emissions were not pulse-like in all PCU treatments during the Chinese cabbage growing season and the average N$_2$O flux was 0.43 mg m$^{-2}$ h$^{-1}$ for the 1PCU. Compared with the 1PCU treatment, the average N$_2$O flux was decreased by 25.58% for the 0.9PCU ($P < 0.05$), 30.23% for the 0.8PCU ($P < 0.05$), and 44.19% for the 0.7PCU ($P < 0.05$).

### Cumulative N$_2$O emissions

Cumulative N$_2$O emissions of CK were at the lowest level (0.47 kg/hm$^2$), which was significantly different from other treatments (Fig 4). Relative to the CU plot (5.82 kg/hm$^2$), N$_2$O emissions were reduced by 32.47% ($P < 0.05$) for the 1PCU,

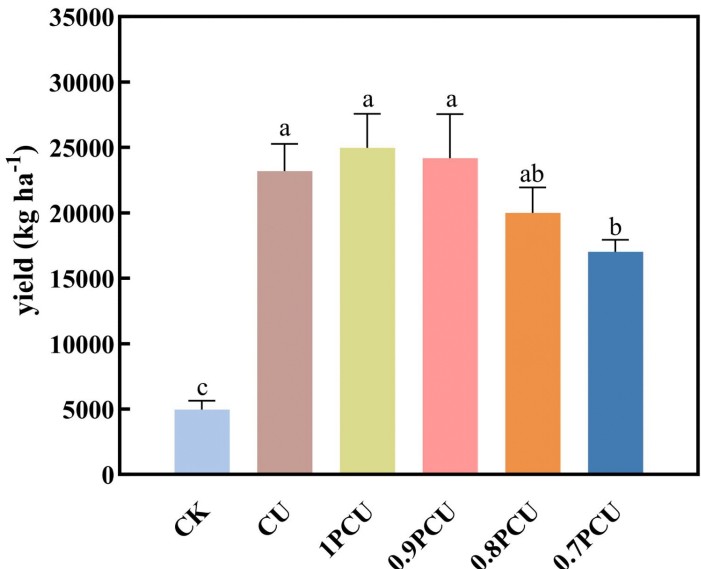

**Fig 2. Variations in the yield of cabbage after the application of different N treatments during the planting of cabbage.** Different lowercase or uppercase letters indicate significant differences at $P < 0.05$.

**Table 1. N uptake and NUE[a] of different N treatments in plots planted with Chinese cabbage N treatment.**

| N treatment | N uptake (kg N ha⁻¹) | NUE (%) |
|---|---|---|
| CK | 90.6 d[b] | — |
| CU | 174.8 bc | 32.4 b |
| 1PCU | 213.1 a | 47.1 a |
| 0.9PCU | 204.6 a | 48.7 a |
| 0.8PCU | 194.4 ab | 49.9 a |
| 0.7PCU | 159.6 c | 32.9 b |

[a]NUE = [N uptake in N-fertilized plots (kg) − N uptake in zero-added N plots (kg)]/ Applied-N rate (kg)×100.
[b]Different lowercase letters indicate significant differences at P < 0.05.

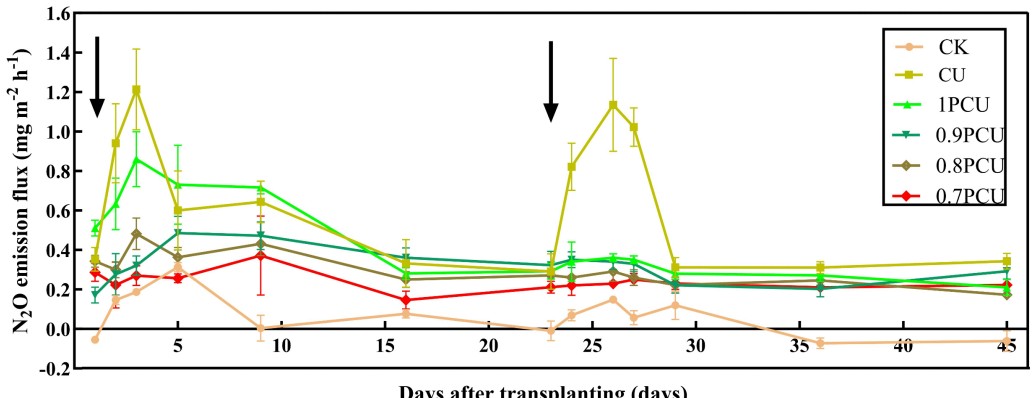

**Fig 3. Changes in N₂O emission flux from soils planted with cabbage under different N management systems.** Arrows indicate dates application of the N fertilizer.

by 45.88% (P < 0.05) for the 0.9PCU, by 47.42% (P < 0.05) for the 0.8PCU, and by 55.67% (P < 0.05) for the 0.7PCU. The N₂O emissions gradually decreased by 2.19–3.24 kg/hm² in the PCU treatment plots as the N application rates were reduced by 0–30%. Compared with the 1PCU treatment, N₂O-N losses were reduced by 13.26% (P > 0.05), by 15.98% (P > 0.05) for the 0.9 PCU and the 0.8 PCU treatments, whereas significantly reduced the discharge amount of N₂O-N by 34.61% for the 0.7 PCU treatment.

### NH₃ volatilization flux

Fig 5 shows the temporal variations in NH₃ volatilization as affected by the different fertilization treatments during the Chinese cabbage growing season. Most losses of NH₃-N arose within 10 days after N fertilization, which initially increased to a peak value and then decreased until emissions ceased. The CU plot showed the highest emission flux around three days after N application, with the peak fluxes reaching 4.06 and 2.47 kg N ha⁻¹ d⁻¹, respectively. Compared with peak fluxes of CU, the peak NH₃ volatilization rates under PCU treatments were delayed by 3–5 days, and the peak ammonia volatilization rates were significantly reduced by 48.28–64.78% and 18.62–42.11% compared to those of the basal fertilizer and topdressing stage under CU treatment, respectively. Dynamic trends of NH₃ volatilization in all PCU treatments were consistent, although flux values were different. Relative to the 1PCU treatment, the average NH₃ flux was significantly reduced by 16.9% for the 0.9PCU, 20.89% for the 0.8PCU, and 27.94% for the 0.7PCU treatments.

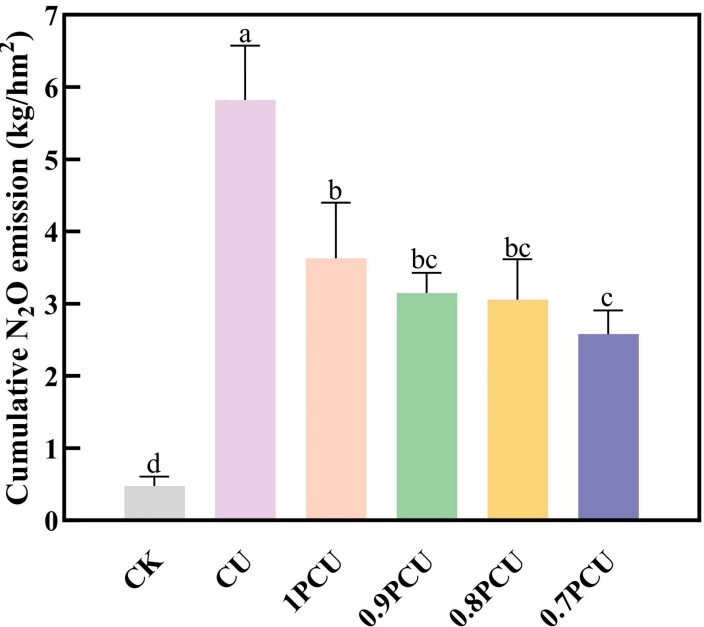

**Fig 4. Variations in N$_2$O emissions from soils planted with Chinese cabbage under different N management systems.** Different lowercase or uppercase letters indicate significant differences at P < 0.05.

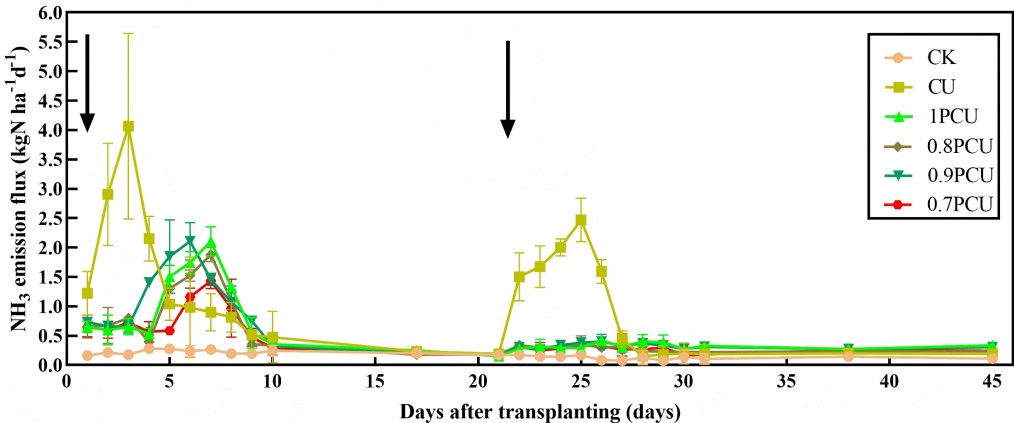

**Fig 5. Changes in NH$_3$ volatilization flux from soils planted with Chinese cabbage under different N management systems.** Arrows indicate dates of fertilizer N application.

## Cumulative NH$_3$ volatilization

There were significant variations in cumulative NH$_3$ volatilization among the different plots with applied N compared to the CK plot (Fig 6). The maximum NH$_3$ emissions were observed in CU (35.24 kg N ha$^{-1}$). Compared to the CU plot, the NH$_3$-N emissions were reduced by 20.74% (P < 0.05) for the 1PCU, 26.99% (P < 0.05) for the 0.9PCU, 30.97% (P < 0.05) for the 0.8PCU and 40.91% (P < 0.05) for the 0.7PCU. NH$_3$-N emissions gradually decreased by 7.32–14.35 kg N ha$^{-1}$ among the PCU-treated plots as the rates of application of N were reduced by 0% to 30%. Relative to 1PCU treatment, cumulative

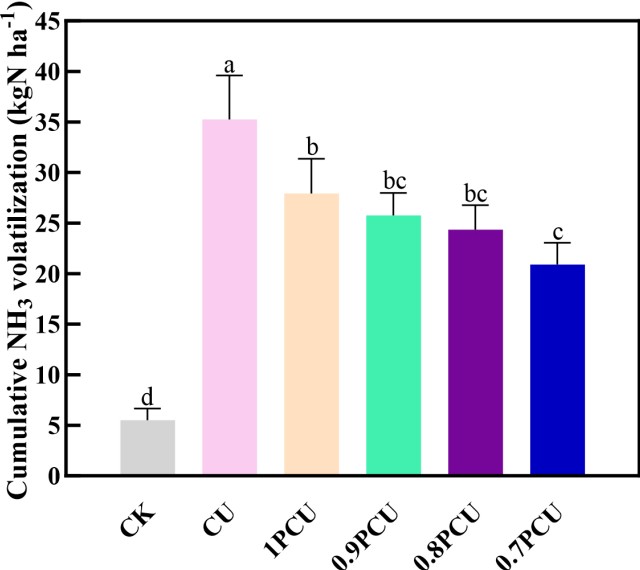

**Fig 6. Variations of $NH_3$ volatilization from soils planted with Chinese cabbage under different N management systems.** Different lowercase or uppercase letters indicate significant differences at $P < 0.05$.

$NH_3$ volatilization was significantly reduced by 25.45% for 0.7PCU but without a significant difference for 0.9PCU and 0.8PCU.

## Emission ratios (ER) of $N_2$O-N and $NH_3$-N

Relative to the CU treatment, $N_2$O ER was significantly minimized by 34.63–44.39% for all PCU treatments (Table 2). The $N_2$O ER of the 0.7PCU treatment (1.14%) was significantly lower than those of the 1PCU (1.34%), the 0.9PCU (1.30%), and the 0.8PCU (1.25%) treatments, whereas there was no significant difference among PCU treatments with a reduction in the application of N rate of 0% to 20%. Compared to the CU treatment, the effect of different PCU-N application on reducing $NH_3$ emission losses was evident by the significantly lower ER of $NH_3$-N of 24.39% ($P < 0.05$) for the 1 PCU, 24.04% ($P < 0.05$) for the 0.9 PCU, 20.53% ($P < 0.05$) for the 0.8PCU and 26.23% ($P < 0.05$) for the 0.7PCU. However, significant differences in $NH_3$ losses were not present among PCU treatments.

**Table 2. Variations in ER of $N_2$O-N and $NH_3$-N in different N treatments.**

| N treatment | ER | |
|---|---|---|
| | $N_2$O-N (%) | $NH_3$-N (%) |
| CK | — | — |
| CU | 2.05a[a] | 11.4a |
| 1PCU | 1.34b | 8.62b |
| 0.9PCU | 1.32b | 8.66b |
| 0.8PCU | 1.25b | 9.06b |
| 0.7PCU | 1.14c | 8.41b |

[a] Different lowercase or uppercase letters indicate significant differences at $P < 0.05$.

**N rates drive trade-offs between yield and NUE and gaseous-N loss**

Fig 7 clearly revealed that nitrogen fertilizer input was the key factor affecting both crop yield and gas emissions. A very strong positive correlation was observed between N application rates and yield ($r = 0.956$, $p < 0.01$), demonstrating the fundamental role of nitrogen fertilizer in enhancing crop yield. However, increased N rates also significantly elevated cumulative $N_2O$ emissions ($r = 0.781$, $p < 0.01$) and cumulative $NH_3$ volatilization ($r = 0.879$, $p < 0.01$), indicating that increased yield comes at higher environmental costs. Notably, there existed a strong synergistic relationship between yield and NUE ($r = 0.876$, $p < 0.01$), while a relatively weak positive correlation appeared between gaseous-N loss (particularly $N_2O$) and NUE ($r = 0.511$, $p < 0.05$). This suggested that optimizing NUE through improved management practices may tend to mitigate gaseous-N emission intensity to some extent when pursuing high yield.

A linear regression model was selected for the relationship between yield and N rates (Formula 8; Table 3). The overall model was significant ($F = 171.957$, $p < 0.001$) and could explain approximately 91.5% of the variation in yield according to the adjusted $R^2$ value. The regression equation is as follows:

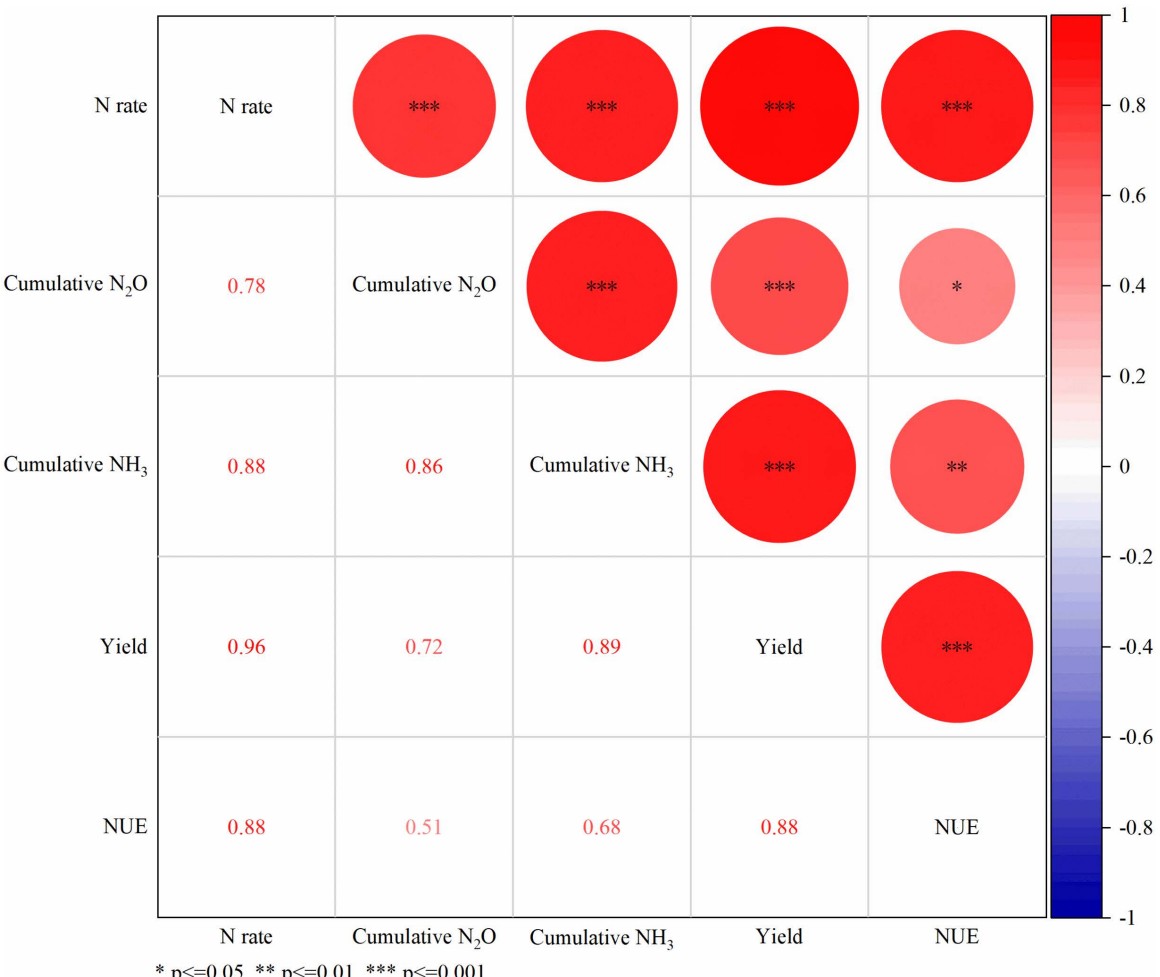

**Fig 7. Correlation analysis between various indicators.** Red represents a positive correlation, and blue represents a negative correlation (* $p \leq 0.05$, ** $p \leq 0.01$, *** $p \leq 0.001$).

**Table 3. Coefficients of Regression Analysis between N rate and yield[a].**

| Model | Regression coefficients | Std. Error | T-values | P-values | VIF |
|---|---|---|---|---|---|
| Constant | 4646.928 | 1213.592 | 3.829 | 0.001 | – |
| N rate | 75.533 | 5.760 | 13.113 | <0.001 | 1.000 |

[a] Dependent Variable: yield

$$y = 4646.928 + 75.533x \tag{8}$$

where $y$ is the yield (kg ha$^{-1}$), $x$ is the N rate (kg ha$^{-1}$).

The response of NUE to N rates was modeled as a quadratic polynomial function (Formula 9; Table 4). To eliminate the collinearity problem of polynomial regression, centralization is adopted. The hypothesis test result of the model was significant ($F = 31.381$, $p < 0.001$), which could explain approximately 79.1% of the variation in NUE based on the adjusted $R^2$ value. The regression equation is as follows:

$$y = 0.501 + 0.001 \times (x - 221) - 5.005 \times 10^{-5} \times (x - 221)^2 \tag{9}$$

where $y$ is the NUE (%), $x$ is the N rate (kg ha$^{-1}$), The average N rate is 221 kg ha$^{-1}$.

Regression statistical analysis revealed both the relationship between N rates and nitrous oxide emission, and that with ammonia volatilization, were exponential (Formulas 10 and 11). Their hypothesis testing was completed by performing a natural logarithmic transformation on the dependent variable and then conducting linear regression analysis with the independent variable (Table 5). Our results showed the overall model was significant for $N_2O$ ($F = 104.529$, $p < 0.001$) and $NH_3$ ($F = 135.100$, $p < 0.001$), proclaiming an excellent fitting effect. Meanwhile, the adjusted $R^2$ value also indicated that the model could explain approximately 85.9% of the variation in $N_2O$ and 88.7% of the variation in $NH_3$. The regression equation is as follows:

$$y = 0.454 \times e^{0.009x} \tag{10}$$

**Table 4. Coefficients of Regression Analysis between N rate and NUE[a].**

| Model | Regression coefficients | Std. Error | T-values | P-values | VIF |
|---|---|---|---|---|---|
| Constant | 0.501 | 0.011 | 46.051 | <0.001 | – |
| N rate-221 | 0.001 | 0.000 | 4.342 | 0.002 | 1.000 |
| (N rate-221)$^2$ | −5.005E-5 | 0.000 | −4.977 | 0.001 | 1.000 |

[a] Dependent Variable: NUE.

**Table 5. Coefficients of Regression Analysis between N rate and ln_N$_2$O (or ln_NH$_3$)[a].**

| Dependent Variable | Model | Regression coefficients | Std. Error | T-values | p-Value | VIF |
|---|---|---|---|---|---|---|
| ln_N$_2$O | Constant | −0.790 | 0.181 | −4.364 | <0.001 | – |
| | N rate | 0.009 | 0.001 | 10.224 | <0.001 | 1.000 |
| ln_NH$_3$ | Constant | 1.694 | 0.124 | 13.698 | <0.001 | – |
| | N rate | 0.007 | 0.001 | 11.623 | <0.001 | 1.000 |

[a] Dependent Variable: ln_N$_2$O or ln_NH$_3$.

where y is the Cumulative $N_2O$ volatilization (kg ha$^{-1}$), x is the N rate (kg ha$^{-1}$).

$$y = 5.442 \times e^{0.007x} \tag{11}$$

where $y$ is the Cumulative $NH_3$ volatilization (kg ha$^{-1}$), $x$ is the N rate (kg ha$^{-1}$).

## Discussion

### Impacts of PCU application on cabbage yield and NUE

Crop yield is primarily governed by genetic factors but is also significantly influenced by environmental conditions and agronomic practices, including nitrogen (N) fertilization and cultivation management [31]. Our results demonstrated that Chinese cabbage yield remained statistically unchanged between CU and PCU plots, even with N reductions of 0–20%, despite higher N uptake and NUE in PCU treatments (Table 1). This aligns with prior studies on open-field vegetable systems in China [21,32], suggesting substantial potential to reduce N inputs without compromising yield while enhancing N utilization efficiency. This divergence likely stems from variations in coating materials and soil temperature, both known to regulate nitrogen release kinetics [18]. The gradual N release from PCU is better synchronized with crop demand, enhancing nutrient uptake and plant growth—consistent with Xie et al. [19]. However, the 26.6% yield reduction in 0.7PCU (182 kg N ha$^{-1}$) versus CU plots reflects insufficient N supply, as evidenced by lower N uptake and NUE (Table 1), ultimately limiting *pakchoi* productivity.

PCU treatments (1PCU-0.8PCU) exhibited significantly higher NUE (47.1–49.9%) compared to conventional urea (CU, 32.4%). This aligns with previous findings where a 20% N reduction (624 kg N ha$^{-1}$) improved NUE from 10.7% to 19.1% in Chinese cabbage [33]. Similarly, Guo et al. [34] reported a 33% N reduction (315 kg N ha$^{-1}$) increased NUE to 15.7–17.0% versus conventional fertilization (4.7–5.6%). The higher NUE values in our study likely reflect differences in environmental conditions, soil characteristics, and agronomic practices [2,35].

Collectively, our findings demonstrate that PCU with 0–20% nitrogen reduction maintained comparable cabbage yields while significantly improving NUE relative to conventional fertilization. However, these single-season results from the subtropical hilly drylands of southern China require further validation through multi-year trials to: (1) verify the synchronization between PCU-N release patterns and crop demand, and (2) assess the long-term consistency of yield responses to varying PCU-N reduction rates.

### The environmental effects of PCU application

Vegetable production is a significant source of gaseous-N emissions in China, wherein nitrogen fertilizer application contributes substantially via emissions of $N_2O$ and $NH_3$ [10]. To address this, controlled-release fertilizers offer a promising mitigation strategy. Despite their well-documented efficacy, the synergistic mechanisms underpinning their mitigation potential remain poorly quantified, particularly within complex topographies such as subtropical hilly drylands [36,37]. Previous studies in China had primarily investigated strategies such as reducing conventional urea application rates [36], evaluating polymer-coated urea (PCU) in temperate cropping systems [37], or partial organic substitution for synthetic fertilizer [38]. Distinct from these approaches, our study introduced a novel strategy that synergistically combined a low N-rate with controlled-release technology, specifically tailored for Chinese cabbage production in the subtropical hilly drylands. This integrated approach is designed to enhance nitrogen use efficiency (NUE) without compromising yield, while also circumventing the potential risk of elevated $N_2O$ emissions associated with organic amendments in high-N-input soils.

Agricultural $N_2O$ emissions, a key nitrogen loss pathway, primarily result from nitrification and denitrification processes driven by fertilizer application [39]. Our study demonstrated that PCU significantly mitigated both $N_2O$ flux and cumulative emissions, a reduction mediated through the controlled release of nitrogen via its semipermeable polymer membrane coating [40]. Contrary to established correlations between $N_2O$ flux and environmental factors [41], our study found no

significant relationship with precipitation or air temperature in PCU-treated plots (Fig 1 and 3). This divergence likely reflects: (1) the enhanced crop nitrogen uptake promoted by PCU, along with favorable environmental conditions, effectively reduces the amount of nitrogen available for loss pathways, thereby mitigating potential environmental emissions [42], and (2) insufficient N supply for surface soil microorganisms [43–44] – potentially explaining the absence of distinct emission peaks. $N_2O$ emissions decreased progressively in response to a 0–30% reduction in PCU-N application rates, demonstrating a clear dose-response relationship (Fig 4; Table 5). The 0.7PCU treatment showed significantly lower $N_2O$ flux and cumulative emissions compared to other PCU treatments (Figs 3-4), though no significant differences were observed among 0–20% reduction rates. Similarly, the 0.7PCU treatment had an emission ratio (ER) that was 8.8–14.9% lower than that of the 10–20% reduction treatments, although ER did not differ significantly among treatments with 0–20% reductions (Table 2). These findings highlighted the Applied-N rate as the critical factor controlling the $N_2O$ mitigation efficacy of PCU, consistent with previous reports [18].

Ammonia volatilization, influenced by soil properties, fertilization practices, and environmental conditions [45], typically peaks within 3–7 days post-application, consistent with previous reports of rapid volatilization completion within nine days [46]. Our results demonstrated that PCU application delayed and reduced $NH_3$ emission peaks compared to conventional urea (CU), likely due to: (1) controlled nitrogen release timing minimizing rainfall-induced losses [47], and (2) reduced nitrogen inputs (10–30% lower in PCU treatments). Cumulative $NH_3$ emissions were significantly lower for all PCU treatments, aligning with findings in maize systems [18,19]. Notably, the 0.7PCU showed the lowest cumulative emissions (Fig 6), suggesting enhanced $NH_3$ mitigation at lower application rates [18], though differences among the 1PCU, the 0.9PCU, and the 0.8PCU were non-significant.

## The economic benefits of PCU application

Despite the agronomic advantages of polymer-coated urea (PCU), its higher cost compared to conventional urea (CU) remains an adoption barrier. Using the integrated assessment method of Xie et al. [19], which incorporates environmental externalities into cost-benefit analysis, we demonstrated that reducing PCU-N application by 0–20% offered an economically viable and sustainable strategy for *pakchoi* production (Table 6). The reason was that compared with CU, a 0–20% reduction in PCU-N application did not significantly decrease the yield economic benefit (YEB), whereas a

Table 6. The economic benefit evaluation of nitrogen fertilizer management of Chinese cabbage cultivation.

| N Treatment | YEB[a] (CNY ha-1) | N input cost[b] (CNY ha-1) | Environmental cost[c] (CNY ha-1) | | NEB[f] (CNY ha-1) | |
|---|---|---|---|---|---|---|
| | | | The lowest[d] | The highest[e] | The lowest[g] | The highest[h] |
| CU | 34795ab[i] | 650 | 1337a | 8830a | 25315ab | 32808ab |
| 1PCU | 37454a | 1170 | 1014ab | 6863ab | 29420a | 35269a |
| 0.9PCU | 36249ab | 1053 | 927b | 6306b | 28891a | 34270ab |
| 0.8PCU | 29985bc | 936 | 880b | 5969b | 23079bc | 28169bc |
| 0.7PCU | 25533c | 819 | 753b | 5115b | 19599c | 23961c |

[a]Yield economic benefit (YEB, CNY ha–1), YEB = Y × r

[b]N input cost (CNY ha–1) = N × p

[c]Environmental cost$=\sum_{a=1}^{b}$ Ca × Pa × m, the environmental cost includes $NH_3$-N and $N_2O$-N.

[d]The lowest denotes the lowest environmental cost of N loss.

[e]The highest denotes the highest environmental cost of N loss.

[f]Net economic benefit (NEB, CNY ha–1), NEB = YEB – N input cost – Environmental cost

[g]The lowest = YEB – N input cost – the highest environmental cost

[h]The highest = YEB – N input cost – the lowest environmental cost

[i]Different letters indicate significant differences at P<0.05.

10% or 20% reduction significantly lowered environmental costs. Although PCU involves higher fertilizer input, its ability to maintain yield and substantially lower environmental externalities led to improved net economic performance. Thus, a 20% reduction in PCU-N application is recommended as a feasible strategy for nitrogen management and achieving ideal benefits.

A critical interpretation of these results demonstrates that the superior net return of common urea is contingent upon highly favorable conditions (i.e., lower-bound environmental cost estimates). Under the more realistic and conservative scenario of upper-bound environmental costs—which represent a greater societal burden—the economic advantage of common urea diminishes substantially, with the net benefit gap closing to within approximately 9% of that achieved by 0.8PCU. This fundamental shift in the cost-benefit calculus underscores a key insight: the marginally higher net income from common urea under optimal conditions fails to offset its significantly greater environmental externalities. This is quantitatively evidenced by a total nitrogen loss of 41.06 kg ha$^{-1}$ for common urea, which is 50% higher than the 27.40 kg ha$^{-1}$ loss from the reduced PCU treatment. Consequently, the minimal economic benefit of common urea cannot rationalize its disproportionately large environmental footprint.

Therefore, from a policy perspective that values sustainability and long-term environmental health, reducing PCU application by 20% represents a feasible and responsible strategy. It achieves a remarkable reduction in nitrogen pollution with a manageable economic trade-off. To bridge the remaining economic gap and encourage farmer adoption, policy instruments such as subsidies for environmentally friendly fertilizers or payments for ecosystem services could be effectively deployed.

## Conclusion

This study evaluated the impact of polymer-coated urea (PCU) application rates on Chinese cabbage yield and nitrogen losses via $NH_3$ volatilization and $N_2O$ emissions. Field observations demonstrated that reducing the PCU-N rate by 0–20% maintained crop yield compared to conventional urea practice, while simultaneously decreasing cumulative $N_2O$ and $NH_3$ emissions by 32.5–47.4% and 20.7–31.0%, respectively. However, a 30% N reduction significantly compromised yield due to nutrient deficiency, despite offering greater emission reductions. Therefore, a 20% reduction in the PCU-N application rate is recommended as the optimal trade-off, sustaining yield while effectively enhancing nitrogen use efficiency and mitigating environmental nitrogen losses. The long-term efficacy of this strategy for continuous cabbage production requires further validation.

## Supporting information

**S1 Fig. Release rate curve in still water at 25°C.**
(TIF)

**S2 Dataset. Original data.**
(XLSX)

## Acknowledgments

We gratefully acknowledge the technical assistance provided by the Liuyang Experimental Station of Hunan Agricultural University.

## Author contributions

**Conceptualization:** Xiangmin Rong.

**Data curation:** Xujiao He.

**Methodology:** Xiangmin Rong.

**Supervision:** Yong Xie.

**Validation:** Li Tang.

**Visualization:** Wei Ming.

**Writing – original draft:** Lian Xie, Yifei Du.

**Writing – review & editing:** Yong Xie.

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
