## [Decision Letter · Decision Letter 0]

18 Jul 2025

Dear Dr. Xie,

Thank you for submitting your manuscript to PLOS ONE. After careful consideration, we feel that it has merit but does not fully meet PLOS ONE’s publication criteria as it currently stands. Therefore, we invite you to submit a revised version of the manuscript that addresses the points raised during the review process.

We look forward to receiving your revised manuscript.

Kind regards,

Paulo H. Pagliari

Academic Editor

PLOS ONE

Journal Requirements:

https://scijournals.onlinelibrary.wiley.com/doi/10.1002/jsfa.9421

https://www.sciencedirect.com/science/article/abs/pii/S0167880923001883?via%3Dihub

In your revision ensure you cite all your sources (including your own works), and quote or rephrase any duplicated text outside the methods section. Further consideration is dependent on these concerns being addressed.

“This study was financially supported by S&T Plan Project of Guizhou Province[ZK�2024�- General Project 029], Guizhou University Doctoral Fund [grant (2022) 35] and Basic Research Projects of Guizhou University[grant(2023) 37].”

Additional Editor Comments:

Please use the reviewer comments to revise your manuscript

Reviewers' comments:

Reviewer's Responses to Questions

**Comments to the Author**

1. Is the manuscript technically sound, and do the data support the conclusions?

Reviewer #1: Yes

2. Has the statistical analysis been performed appropriately and rigorously?

Reviewer #1: Yes

3. Have the authors made all data underlying the findings in their manuscript fully available?

Reviewer #1: Yes

4. Is the manuscript presented in an intelligible fashion and written in standard English?

Reviewer #1: No

Reviewer #1: This field study quantifies how polymer-coated urea (PCU) with 0–30 % reduced N application affects the yield, N-use efficiency (NUE), N2O and NH3 losses of Chinese cabbage. The topic is highly relevant to sustainable vegetable production in China. The experiment is well structured, the data set is complete, and the main message (a 20 % PCU-N reduction maintains yield and cuts gaseous N losses by ~50 %) is clear and policy-relevant. However, several methodological, statistical and mechanistic issues need to be addressed before the paper can be accepted.

1. The PCU is described only as “polyurethane-coated, 40-day release”. Add specific information about PCU characterization such as coating rate (% w/w), membrane thickness, and ideally a cumulative release curve at 25°C (or cite the manufacturer’s technical sheet). Without these data it is hard to reproduce the work or compare with other PCU types and effects.

2. Include regression equations (exponential or linear-plateau) between N rate and cumulative N2O, NH3, yield, NUE. This will strengthen the dose-response discussion.

3. The Discussion proposes three mechanisms for reduced N2O (slow release, physical barrier, prolonged NH4+ dominance). These remain speculative without inorganic-N data. Add at least three sampling dates for NH4+-N and NO3–-N (0–10 cm) or remove the unsupported claims.

4. Include 2–3 recent open-field vegetable PCU studies to highlight novelty in the Discussion.

5. The price of PCU is higher than that of ordinary urea, and it is necessary to give input-output comparisons (fertilizer costs, yield changes, and environmental externalities) to enhance policy operability. A short paragraph is sufficient.

6. The time in the title of Fig 1 is "2022", which is inconsistent with the experimental year 2024 in the article.

7. Check the English grammar throughout (such as Line 191: “it had no effect cabbage yield” missing preposition “on”).

After addressing the above points, this manuscript will make a solid contribution to the literature on precision N management in vegetable systems.

**Do you want your identity to be public for this peer review?** For information about this choice, including consent withdrawal, please see our Privacy Policy

Reviewer #1: **Yes: ** Hanqing Wu

---

## [Author Response · Author response to Decision Letter 1]

31 Aug 2025

Thank you for your time and constructive comments. We have carefully revised our manuscript according to the reviewers' suggestions. We believe the manuscript has been significantly improved and hope it now meets the journal's standards.

---

## [Decision Letter · Decision Letter 1]

11 Sep 2025

Dear Dr. Xie,

Thank you for submitting your manuscript to PLOS ONE. After careful consideration, we feel that it has merit but does not fully meet PLOS ONE’s publication criteria as it currently stands. Therefore, we invite you to submit a revised version of the manuscript that addresses the points raised during the review process.

We look forward to receiving your revised manuscript.

Kind regards,

Paulo H. Pagliari

Academic Editor

PLOS ONE

Journal Requirements:

Reviewer's Responses to Questions

**Comments to the Author**

Reviewer #1: All comments have been addressed

2. Is the manuscript technically sound, and do the data support the conclusions?

Reviewer #1: Yes

3. Has the statistical analysis been performed appropriately and rigorously?

Reviewer #1: Yes

4. Have the authors made all data underlying the findings in their manuscript fully available?

Reviewer #1: Yes

5. Is the manuscript presented in an intelligible fashion and written in standard English?

Reviewer #1: Yes

Reviewer #1: The authors have responded positively to most of my previous comments. Nevertheless, two minor issues remain. I recommend accepting it after modification.

1. The cost-benefit section uses “established unit costs” without citation. Please provide reference for the social cost of N2O-N, NH3-N (USD or CNY kg-1) and specific calculation methods and processes.

2. The discussion still speculates that PCU limits substrate supply to reduce N2O, but there is soil NH4+ and NO3− data. Please delete this speculation or refer to direct evidence of similar soils.

**Do you want your identity to be public for this peer review?** For information about this choice, including consent withdrawal, please see our Privacy Policy

Reviewer #1: **Yes: ** Hanqing Wu

---

## [Author Response · Author response to Decision Letter 2]

21 Sep 2025

We appreciate the reviewers' insightful feedback. We have addressed all the points raised and provided a point-by-point response to each comment in the response letter. We hope the revisions are satisfactory.

---

## [Decision Letter · Decision Letter 2]

7 Oct 2025

Dear Dr. Xie,

Thank you for submitting your manuscript to PLOS ONE. After careful consideration, we feel that it has merit but does not fully meet PLOS ONE’s publication criteria as it currently stands. Therefore, we invite you to submit a revised version of the manuscript that addresses the points raised during the review process.

We look forward to receiving your revised manuscript.

Kind regards,

Paulo H. Pagliari

Academic Editor

PLOS ONE

Journal Requirements:

Reviewers' comments:

Reviewer's Responses to Questions

**Comments to the Author**

Reviewer #1: All comments have been addressed

2. Is the manuscript technically sound, and do the data support the conclusions?

Reviewer #1: Yes

3. Has the statistical analysis been performed appropriately and rigorously?

Reviewer #1: Yes

4. Have the authors made all data underlying the findings in their manuscript fully available?

Reviewer #1: Yes

5. Is the manuscript presented in an intelligible fashion and written in standard English?

Reviewer #1: Yes

Reviewer #1: The authors have satisfactorily addressed the scientific criticisms raised in Round 2. The economic analysis now includes proper citations for the unit social costs of N2O-N and NH3-N, and the speculative mechanistic sentences have been removed/re-phrased. Nevertheless, several formal issues remain that must be cleared before the paper can be accepted. I recommend Minor Revision; if all items below are fulfilled in one go, no further external review will be necessary.

Many language and typographical errors. For example:

Line 202: “Environment benefi” → “Environmental cost”

Line 477: “The higest” → “The highest”

Please run a spell-checker or professional editing service once more.

**Do you want your identity to be public for this peer review?** For information about this choice, including consent withdrawal, please see our Privacy Policy

Reviewer #1: **Yes: ** Hanqing Wu

---

## [Author Response · Author response to Decision Letter 3]

11 Oct 2025

We sincerely thank the reviewer for their positive feedback on our revisions. To address the remaining formal issues, we have meticulously polished the Abstract, Methods, and Conclusions sections for better clarity and flow.

---

## [Decision Letter · Decision Letter 3]

6 Nov 2025

Polymer-coated urea effects on yield and gaseous loss of nitrogen for planting Chinese cabbage

PONE-D-25-22820R3

Dear Dr. Xie,

We’re pleased to inform you that your manuscript has been judged scientifically suitable for publication and will be formally accepted for publication once it meets all outstanding technical requirements.

Kind regards,

Paulo H. Pagliari

Academic Editor

PLOS ONE

Additional Editor Comments (optional):

Reviewers' comments:

Reviewer's Responses to Questions

**Comments to the Author**

Reviewer #1: All comments have been addressed

2. Is the manuscript technically sound, and do the data support the conclusions?

Reviewer #1: Yes

3. Has the statistical analysis been performed appropriately and rigorously?

Reviewer #1: Yes

4. Have the authors made all data underlying the findings in their manuscript fully available?

Reviewer #1: Yes

5. Is the manuscript presented in an intelligible fashion and written in standard English?

Reviewer #1: Yes

Reviewer #1: The manuscript has been substantially improved. All essential technical issues raised in the previous rounds have been adequately addressed. I recommend acceptance in its present form.

**Do you want your identity to be public for this peer review?** For information about this choice, including consent withdrawal, please see our Privacy Policy

Reviewer #1: **Yes: ** Hanqing Wu

---

## [Editor Report · Acceptance letter]

PONE-D-25-22820R3

PLOS ONE

Dear Dr. Xie,

I'm pleased to inform you that your manuscript has been deemed suitable for publication in PLOS ONE. Congratulations! Your manuscript is now being handed over to our production team.

Kind regards,

on behalf of

Dr. Paulo H. Pagliari

Academic Editor

PLOS ONE